

# Individual characteristics, including olfactory efficiency, age, body mass index, smoking and the sex hormones status, and food preferences of women in Poland

Magdalena Hartman-Petrycka[1], Joanna Witkoś[2], Agata Lebiedowska[1] and Barbara Błońska-Fajfrowska[1]

[1] Department of Basic Biomedical Science, Faculty of Pharmaceutical Sciences in Sosnowiec, Medical University of Silesia, Katowice, Poland
[2] Medicine and Health Science, Andrzej Frycz Modrzewski Krakow University, Kraków, Poland

## ABSTRACT

**Background.** Food choices made by most people mainly depend on food preferences. Knowing how certain factors affect food preferences can help dietitians working with women to understand the relationship between individual factors and the challenges faced by the women in changing eating habits. The aim of the study was to examine the food preferences of women and to assess the impact of the sense of smell, age, body mass index (BMI), smoking and hormonal status (phase of the menstrual cycle, hormonal contraception) on the declared pleasure derived from eating various types of food.

**Methods.** A total of 190 women living in the Górnośląsko-Zagłębiowska Metropolis in Poland aged 18–75 (19.29–26.71 RNO) years participated in the study. The collected survey data included age, BMI, smoking, phase of the menstrual cycle and hormonal contraception. Olfactory sensitivity was measured by T08 olfactometer. Additionally, food preferences were assessed, using 24 different food types, which were presented as pictures. To evaluate food preferences 10 cm visual analogue scale was used.

**Results.** The most liked foods were: fruits (M = 8.81, SD = 1.67), sweet desserts (M = 8.44, SD = 2.30), vegetables and salads (M = 8.08, SD = 2.24), chocolate (M = 7.84, SD = 2.76), and poultry (M = 7.30, SD = 2.47). The least liked foods were: salty products (M = 4.98, SD = 3.03), milk soup (M = 3.30, SD = 3.13), and seafood (M = 2.99, SD = 3.28). The influence of the analyzed factors on the degree of liking six food types was shown. Women with better ability to name scents preferred sausages/ham and beef/pork. Women with a higher BMI had lower preference for jellybeans and broth. Women who were heavier smokers had lower preference for milk soup. In women using hormonal contraception, pleasure from eating sausages and ham was higher than compared to women in all phases of the monthly cycle. In women in the follicular phase the pleasure from eating pasta was lower when compared to women in the ovulatory phase, the luteal phase and those using hormonal contraception. In women in the ovulatory phase the pleasure from eating candies and jellybeans was lower when compared to women in the follicular phase, the luteal phase and those using hormonal contraception. In women in the ovulatory phase, also pleasure from

Corresponding author
Joanna Witkoś, jwitkos@afm.edu.pl

eating broth was lower when compared to women in the luteal phase and those using hormonal contraception.

**Conclusions**. Among women in Poland, the top five preferred food types are fruits, sweet desserts, vegetables/salads, chocolate and poultry. To confirm the extent to which the declared pleasure derived from eating these food types translates into health condition, further research on the consumption of these food types is necessary. The impact of the sense of smell, BMI, smoking, or menstrual cycle phase and hormonal contraception on the declared pleasure derived from eating was observed for six out of twenty-four food types. The hormonal status was the factor most significantly influencing food preferences.

## INTRODUCTION

Today, the prevalence of overweight and obesity is considered a global pandemic. The problem is so serious that, in 2013, the World Health Organization members set the goal of halting the obesity pandemic by 2025 (*World Health Organization, 2013*). However, differences between regions are observed in this respect, with the problem of excess body weight being more severe in some of them, and less in the others. Studies conducted by *Berghöfer et al. (2008)*, shown that the prevalence of obesity in Central and Eastern Europe is higher compared to Western and Northern countries.

The prevalence of overweight and obesity is determined on a basis of the body mass index (BMI). In a collective analysis conducted by the Global Burden of Disease Study, it was found that the BMI increase above 25 was associated with a higher risk of certain diseases, including cardiovascular diseases, type two diabetes, some cancers, osteoarthritis, and kidney disease. Overweight and obesity are also factors exacerbating certain conditions, such as respiratory system diseases and some types of cancers, *e.g.*, colorectal cancer (*Whitlock et al., 2009*). Additionally, obesity contributes to hypertension, glucose intolerance, dyslipidemia, and chronic inflammation (*Stepniak et al., 2016*).

According to the European Health Interview Survey (EHIS) in 2014 in Poland, the percentage of obese and overweight people was 16.7% and 36.6%, respectively, of the population aged 15 and over (*Statistics Poland, 2016*). This result is higher than the average for the 28 European Union countries, which was 15.4% and 34.8% for obese and overweight people, respectively (*Zgliczyński, 2017*). Overall, the percentage of people with excess body weight is rising (*Thom & Lean, 2017*).

The other extreme in improper nutrition is malnutrition and deficiency of individual micro and macro elements in the diet, which can cause a number of health complications, *e.g.*, iron deficiency anemia, neurological disorders due to vitamin B12 deficiency, and many others. Specific nutrition standards were developed for the Polish population (*Przygoda et al., 2019*) and specified in the European Union regulations (*European Parliament, 2011*),

regarding the determination of product nutrient levels. However, for most people, eating is a natural daily activity and is based on eating habits.

The formation of eating habits and food preferences, as well as a specific lifestyle (active, sedentary) depend on many factors and numerous interactions between them. A health-promoting attitude is formed in a person from an early age. A child repeats the patterns it observes in its parents, including their preferences and needs for sport activity or healthy eating habits (*Thomson et al., 2020*). Research (*Van Rongen et al., 2020*) showed that the respondents environment had a strong influence on their behavior and choices in the family, circle of friends, or neighborhood. Positive patterns related to the consumption of certain types of food were associated with their consumption as a 'social norm'. The results of many studies (*Simmonds et al., 2016*; *Evensen et al., 2016*) showed that about half of obese children are also obese in adulthood, which, in turn, increases their risk of numerous diseases in adulthood. While childhood patterns are very important, they are not a sole factor influencing food preferences.

This study analyzed the factors that may determine food preferences in adult women. It is known that the choice and consumption of food depends both on the chemosensory properties of products and on the efficiency of the chemical senses responsible for stimulus perception. The publication by *Boesveldt et al. (2017)*, showed that the sense of smell plays an initiating role in eating behavior. The exposure to a smell (orthonasal) increases the appetite. Some reports demonstrated that age-related olfactory disorders change eating behaviors of the studied women (*Gopinath et al., 2016*). It has also been shown that obese people have a reduced ability to identify flavors in retronasal measurements when compared to subjects with normal body weight (*Nettore et al., 2020*). Furthermore, women's physiology changes during the menstrual cycle.

The status of sex hormones and their influence on the women physiology also change during the use of contraceptives and the menopause period. Studies by *Kammoun et al., (2017)* indicate significant changes in caloric intake and body weight during the menstrual cycle. The diet is also affected by both smoking and quitting smoking (*Chao et al., 2017*). The research presented above shows possible directions of research on a relationship between women's individual characteristics and their food preferences. The literature on nutrition uses a wide spectrum of research methods; however, the study of many nutrition aspects and the factors affecting them is extremely complicated, and papers necessarily describe a relatively narrow aspect of the topic. In the methodology of this work, we decided to approach the issue holistically and use a wide range of foods, varied in terms of taste, texture, origin and degree of processing, representing staple foods for women in Poland. We also wanted to assess the influence of the olfactory efficiency in women, simultaneously with additional factors that may modify food preferences.

Finally, the aim of this study was to analyze factors determining specific food preferences in women in the Górnośląsko-Zagłębiowska Metropolis in Poland. The specific objectives included a comprehensive assessment of the declared pleasure derived from eating 24 types of foods well-known to women in Poland, some considered less healthy and with high energy content, and some considered healthy, such as vegetables, fruit and fish. The detailed objectives also included an assessment of factors such as the sense of smell, age,
BMI, smoking, and women's hormonal status (menopause, hormonal contraception, phase of the menstrual cycle), which influence the declared pleasure derived from eating various types of food.

This knowledge may help to identify factors predisposing women to a greater risk of developing diseases of civilization due to their food preferences, and to target those women with educational and preventive projects. Knowing how certain factors affect food preferences can also help dietitians working with patients to understand the relationship between individual factors and the challenges faced by the patient in changing their eating habits.

## MATERIALS & METHODS

### Participants

The study involved 190 women, aged 18–75 years, who were able to understand the research method and to give an informed consent to participation in the project. After outliers were eliminated, the age was 19.29–26.71 years (Table 1). Because the olfactometric test was carried out in the olfactometric laboratory located in Sosnowiec, most of the study participants came from the city of Sosnowiec and the surrounding cities of the Górnośląsko-Zagłębiowska Metropolis in the Silesian Voivodeship, Poland. Volunteers for the study were recruited from among students and employees of the Faculty of Pharmaceutical Sciences in Sosnowiec. The students also recruited women from among their family and friends as a part of the activities of the Faculty research club. An estimated half of the surveyed women studied medicine (Medical Analytics, Pharmacy, Cosmetology, or Biotechnology). No transgender women participated in the study.

Women with no symptoms of an upper respiratory tract infection were qualified for the study. Their nasal patency and symmetry of air flow through the nostrils were confirmed (the total air flow through the nose in a rhinomanometric examination exceeded 280 cm$^3$, the air flow ratio between the right and left nostril ranged from 0.5 to 1.5). A transient nasal obstruction could excessively affect the results of the olfactory efficiency test, so the subjects with obstructed airflow could be classified as anosmic on the test day.

### Volunteers' preparation for the study

The volunteers were asked to avoid consuming spicy foods with intense odors, such as garlic or raw onions, on a day preceding the study. They were also asked to refrain from using strong smelling body care products. The volunteers entered the olfactometric laboratory 30 min before the study in order to adapt to the room conditions. During this time, according to the PN-EN 13725: 2007 standard guidelines, the volunteers were not allowed to eat or drink (other than still water), chew gum, smoke cigarettes, use fragrant cosmetics, nor engage in physical activity. During this time, the rhinomanometric measurement and a health interview were conducted, and anthropometric data and information on the women's monthly cycle were collected. During the health interview, the respondents were asked to answer the following questions in writing: (1) Date of examination, (2) Date of birth, (3) Body weight, (4) Height, (5) Do you have a sore throat? (6) Do you have a headache? (7) Do you have a cold? (8) Have you smoked cigarettes in the past?

**Table 1  Characteristics of participants.**

|  | Mean | Med | SD | Min | Max | MAD | RNO | NRem |
|---|---|---|---|---|---|---|---|---|
| Age (years) | 28.06 | 23 | 12.28 | 18 | 75 | 1.48 | 19.29–26.71 | 46 |
| BMI | 22.46 | 21.3 | 4.15 | 16.65 | 36.73 | 2.85 | 14.18–28.42 | 21 |
| Number of smoking years | 1.82 | 0 | 5.54 | 0 | 33 | 0 | – | – |
| Number of cigarettes per day[*] | 1.32 | 0 | 4.53 | 0 | 50 | 0 | – | – |
| The severity of addiction (pack-years) | 0.82 | 0 | 4.68 | 0 | 60 | 0 | – | 1 |
| N-butanol dilution step (1st step is the most diluted sample) | 6.75 | 7.00 | 3.19 | 1.00 | 16.00 | 2.97 | 0.41–14.41 | 6 |
| Identification test of smell | 4.06 | 4 | 1.08 | 0 | 5 | 1.48 | 0.29–7.71 | 1 |

Notes.

*In the case of three variables: Number of smoking years, Number of cigarettes per day, and The severity of addiction (pack-years), the median was zero (more than 50% cases had the result '0'). Therefore, the MAD also equals zero and it is not possible to calculate the cut-points for detecting outliers. On of these variables: The severity of addiction (pack-years) was included in regression analyses. One clear outlier was removed: a case with the result '60' - the next nearest result was 12.5.

Med, Median; SD, Standard Deviation; Min, Minimum; Max, Maximum; MAD, Median absolute deviation; RNO, Range of results which are not outliers; NRem, Number of cases removed as outliers.

(9) Do you smoke cigarettes? If so, how many years have you been smoking cigarettes? How many cigarettes do you smoke a day? (10) Are you in the menopause? (11) Are you using hormonal contraception? (12) Do you menstruate regularly? (13) Provide an average length of your menstrual cycle. (14) What day of the menstrual cycle are you on?

## Food preferences study

Food preferences were assessed using an album containing photos of 24 types of food: (1. Fish dishes, 2. Egg dishes, 3. Sweet desserts, 4. Chocolate, 5. Candies and jellybeans, 6. Crisps, 7. Dumplings, 8. Pasta, 9. Milk soup (a sweet dish made by pouring hot milk over boiled rice, pasta, oatmeal, chocolate chips, corn flakes and similar products), 10. Milk drinks, 11. Cheese, 12. Vegetables and salads, 13. Fruit, 14. Sausages and ham, 15. Beef and pork, 16. Poultry, 17. Bread, 18. Fast food, 19. Salty products, 20. Sour products, 21. Broth (served the Polish way, with noodles), 22. Soups, 23. Spicy dishes, 24. Seafood. The products were selected to be well known in Poland. Food types were chosen to ensure the use of a wide range of products with different predominant flavors (sweet, salty, sour, spicy), of different food groups (meat, vegetables, fruit, dairy products), and at various levels of processing (fast food, raw vegetables). The photos of the food types used in the study are provided in supplementary materials (Supplement 1). After viewing photos of certain food types, each subject was asked to specify their preference for that food by drawing a vertical line on the linear scale in the "Food Preference Assessment Sheet". The 10 cm visual analogue scale was labelled "0 - no pleasantness" at one end and "10 - maximum pleasantness" at the other (Supplement 2). Pictures containing a specific category of food products usually showed three or more food types. The women had to choose the one which they preferred the most and define their preferences concerning the food type. For example, the fish category included pictures of both baked fish and marinated herring, so it was necessary to specify how pleasant the participant found the most liked of these two fish dishes.

## Olfactory sensitivity test

The olfactory sensitivity tests were performed using gaseous odor samples. Individual olfactory sensitivity thresholds were determined by the dynamic olfactometry method using the Olfactometer T08 by ECOMA, in the olfactometric laboratory, in accordance with the PN-EN 13725: 2007. Undiluted N-butanol at a concentration of 59.9 ppm and the air used to dilute substances were used for the tests. In order to determine their olfactory sensitivity thresholds, the participants were exposed to n-butanol at a series of dilutions, from $2^{16}$ (65536): 0.0009 ppm –step 1, $2^{15}$ (32768): 0.0018 ppm –step 2, $2^{14}$ (16384): 0.0036 ppm –step 3,..., $2^4$ (16): 3.7375 ppm –step 13, $2^3$ (8): 7.4750 ppm –step 14, $2^2$ (4): 14.9500 ppm –step 15, to 0: 0 ppm –step 16. The diluted odor samples were administered to the nose area at a speed of 0.2 m/s, for 2.2 s, and were alternated with the clean air. The subject was not aware of the odorant type or concentration. The respondents had to answer a question: "Can you smell anything ?". A positive answer was signaled by pressing a button next to the olfactometer. After the olfactory sensitivity threshold to n-butanol was determined twice, there was a 15-minute break, and then a match test was performed for five odors (menthol –mint odor, limonene - citrus odor, phenylethyl alcohol - floral odor, eugenol - clove odor, n-butanol - chemical odor, alcohol). The odors were applied onto two cm perfume test strips. The participants sniffed the samples one by one, and then entered the odor name in the answer sheet. The subjects could freely name the odors. All answers that were similar to those presented above were considered correct, *e.g.*, limonene - 'lemon', 'citrus',' 7-up ', 'limonene' or 'lemon scented cleaning product'. The result of the test was the number of correct answers.

The research project was approved by the Bioethics Committee of the Medical University of Silesia (KNW/0022/KB1/47/12). The study was conducted in accordance with the Helsinki Declaration. Every participant provided a written consent after being informed about the study aim, protocol and methodology.

## Statistical analysis

For linear multiple regression, a priori power analysis was performed in order to determine the sample size necessary to observe small, medium, and large effects for six predictors. The analyses were performed using the G*Power 3.1.9.6 application (*Faul et al., 2007*; *Faul et al., 2009*). Two-tailed analyses were performed; and a confidence of 95% was assumed. The sample size was analyzed using the effect size (Cohen $f$) of 0.14, 0.39, and 0.59, which correspond to $f^2$ of 0.02, 0.15, and 0.35 required by G*Power, respectively. The sample sizes required to detect small, medium, and large effects were 652, 89, and 40, respectively. Taking into account the available resources, a sample of 190 participants was assumed sufficient, ensuring the excellent power to detect large and medium effects, but not small ones.

The results were recorded in a Microsoft Excel 2007 spreadsheet, and a statistical analysis was performed using SPSS 21 software. Descriptive statistics were used and a regression model (multiple regression analysis with a nominal (categorical) variable)) was constructed for each tested food type. The dependent variables were pleasantness ratings for each of food types: 1. Fish dishes, 2. Egg dishes, 3. Sweet desserts, 4. Chocolate, 5. Candies and

jellybeans, 6. Crisps, 7. Dumplings, 8. Pasta, 9. Milk soup, 10. Milk drinks, 11. Cheese, 12. Vegetables and salads, 13. Fruit, 14. Sausages and ham, 15. Beef and pork, 16. Poultry, 17. Bread , 18. Fast food, 19. Salty products, 20. Sour products, 21. Broth, 22. Soups, 23. Spicy dishes, 24. Seafood.

The five continuous predictor variables used were the subject's age, BMI, pack-years, olfactory sensitivity threshold, and ability to name odors. The single categorical predictor variable was women's hormonal status of (menopause, contraception, follicular phase, ovulatory phase, or luteal phase). The age was calculated by subtracting the year of birth from the year of the survey. The body mass index was calculated according to the formula: body weight [kg]/height [m]2 on a basis of the data from the questionnaires. For smoking, the frequency indicator of ''pack-years'' was used (a number of cigarette packs smoked per day multiplied by the number of years of smoking). The sex hormones status was determined from direct answers concerning the menopause and the use of hormonal contraception, provided by women in the questionnaires. The phases of the menstrual cycle were calculated assuming that the luteal phase, which length is the most constant of all phases, covers the last 12 days of the cycle, the ovulatory phase usually lasts between 13 and 15 days from the end of the cycle, and the follicular phase begins with bleeding and ends before the beginning of the ovulatory phase.

Assumptions of normality, a lack of homoscedasticity, and a lack of multicollinearity were verified. Normality of residuals was checked by the normal probability plots. Homoscedasticity was analyzed with scatterplots of standardized residuals plotted against standardized predicted values. Multicollinearity was calculated using the Variance Inflation Factor (VIF). It was assumed that VIF below 2.00 (that is, tolerance above 0.5) indicates no multicollinearity.

The assumptions of a lack of multicollinearity and a lack of homoscedasticity were met in all analyses. The assumption of normality of the residuals distribution was reasonably met on all analyses, except for candies and jellybeans, broth, and milk soups. In the analysis for Fast food, the lack of homoscedasticity was questionable. Therefore, in the case of these four analyses, a certain caution is recommended.

To detect and eliminate outliers, the technique based on the median absolute deviation (MAD) was applied, as recommended by *Leys et al. (2013)*. In this method, the median is detracted from each value, and the median for the absolute value of the resulting variable is calculated. This median is then multiplied by the value of 1.4826, giving the MAD value. As recommended by *Leys et al. (2013)*, the MAD is then multiplied by 2.5, and the value is subtracted and added to the median of the raw variable, which generates the range of cases which are not outliers.

Unstandardized regression coefficients (B) are provided for numerical and dichotomous variable predictors, indicating the number of units by which the dependent variable changes when the predictor changes by one, for the 95% confidence interval. In the case of the qualitative factor (hormonal status), the significance of differences between the groups was indicated. For each analysis, the value of the multiple determination coefficient ($R^2$) was also given. Additionally, effect-size indicators were given for each predictor variable, expressed as eta-squared. It was neither useful nor possible to use all indicators from

the database, *e.g.*, smoking now strongly correlates with smoking in the past; pack-years strongly correlate with the number of smoking years and the number of cigarettes smoked per day, etc. The t Statistic was provided for the continuous predictors, and the F statistic (Snedecor-Fisher F) was provided for the categorical predictors, with respective dfs. The Sidak correction was applied for multiple group comparisons.

# RESULTS

Of the 190 women studied, 26 (13.68%) smoked cigarettes. The subjects' hormonal status was as follows: 19 (10.0%) women were menopausal, 44 (24.4%) women used oral contraception, 54 (30.0%) women were at the follicular phase, 10 (7.2%) women had ovulation, and 51 (28.3%) women were at the luteal phase. Further characteristics are summarized in Table 1.

The women declared the greatest pleasure from eating fruit ($M = 8.81$, SD = 1.67), sweet desserts ($M = 8.44$, SD = 2.30), vegetables and salads ($M = 8.08$, SD = 2.24), chocolate ($M = 7.84$, SD = 2.76), and poultry ($M = 7.30$, SD = 2.47) (Table 2). The least liked foods were fast food ($M = 5.46$, SD = 3.40), spicy foods ($M = 5.12$, SD = 3.26), salty products ($M = 4.98$, SD = 3.03), milk soups ($M = 3.30$, SD = 3.13), and seafood ($M = 2.99$, SD = 3.28). In six out of 24 cases, it was possible to build regression models, in which at least one predictor variable statistically significantly explained the variability in women's food preferences (Table 3). The full regression model explained 15% of the candies and jellybeans variance ($R^2_C = 0.15$), and BMI was a significant negative predictor variable ($p = 0.027$, et $a^2 = 0.04$). The women's hormonal status also was a significant predictor variable ($p = 0.022$, et $a^2 = 0.08$). The estimated marginal means of preference for candies and jellybeans in women with a different hormonal status, along with the intergroup differences, are presented in Table 3, Fig. 1. The model for broth preferences explained 13% of the variance, and BMI was a significant negative predictor variable ($p = 0.010$, et $a^2 = 0.05$). The women's hormonal status also was a significant predictor variable ($p = 0.035$, et $a^2 = 0.07$) (Table 3, Fig. 1). Variances in the sausages and ham preferences were explained in 12%, and a significant positive predictor variable was the ability to name scents ($p = 0.013$, et $a^2 = 0.07$). The women's hormonal status also was a significant predictor variable ($p = 0.014$, et $a^2 = 0.10$) (Table 3, Fig. 1). Variances in the pasta preferences were explained in 10% and the women's hormonal status was a significant predictor variable ($p = 0.013$, et $a^2 = 0.09$) (Table 3, Fig. 1). Variations in the beef and pork preferences were explained in 8%, and a significant positive predictor variable was the ability to name scents ($p = 0.046$, et $a^2 = 0.03$). Variations in the milk soups preferences were explained in 6%, and pack-years were a significant negative predictor variable ($p = 0.039$, et $a^2 = 0.03$).

For other types of food: sweet desserts, vegetables and salads, chocolate, poultry, dumplings, egg dishes, cheese, soups, fast food, spicy dishes, salty products, seafood, sour products, fruits, bread, fish dishes, crisps and milk drinks, it was not possible to identify any statistically significant factors (age, BMI, pack-years, olfactory sensitivity, ability to name odors, and hormonal status) that affected food preferences (Table 4).

**Table 2  Ranking of foods.** The value of the womens declared pleasure they derived from eating various types of two dishes, starting with the favourite ones.

|  | Dish | Mean | Med | SD | Min | Max | MAD | RNO | NRem |
|---|---|---|---|---|---|---|---|---|---|
| 1 | Fruit | 8.81 | 9.65 | 1.67 | 2 | 10 | 0.52 | 8.35–10.95 | 56 |
| 2 | Sweet desserts | 8.44 | 9.50 | 2.30 | 0 | 10 | 0.74 | 7.65–11.35 | 47 |
| 3 | Vegetables and salads | 8.08 | 8.80 | 2.24 | 0.5 | 10 | 1.78 | 4.35–13.25 | 18 |
| 4 | Chocolate | 7.84 | 9.00 | 2.76 | 0 | 10 | 1.48 | 5.29–12.71 | 35 |
| 5 | Poultry | 7.30 | 7.80 | 2.47 | 0 | 10 | 2.52 | 1.50–14.10 | 7 |
| 6 | Bread | 7.18 | 7.50 | 2.24 | 0.2 | 10 | 2.67 | 0.83–14.17 | 1 |
| 7 | Pasta | 7.13 | 7.70 | 2.46 | 0.9 | 10 | 2.89 | 0.47–14.93 | 1 |
| 8 | Dumplings | 6.88 | 7.20 | 2.62 | 0 | 10 | 3.26 | −0.95–15.35 | 0 |
| 9 | Egg dishes | 6.78 | 7.40 | 2.72 | 0 | 10 | 2.89 | 0.17–14.63 | 1 |
| 10 | Cheese | 6.57 | 7.00 | 2.83 | 0 | 10 | 2.97 | −0.41–14.41 | 0 |
| 11 | Soups | 6.56 | 6.85 | 2.71 | 0 | 10 | 2.97 | −0.56–14.26 | 0 |
| 12 | Fish dishes | 6.40 | 6.50 | 2.79 | 0 | 10 | 2.97 | −0.91–13.91 | 0 |
| 13 | Candies and jellybeans | 6.34 | 7.00 | 3.23 | 0 | 10 | 4.30 | −3.75–17.75 | 0 |
| 14 | Broth[*] | 6.21 | 7.00 | 3.32 | 0 | 10 | 4.23 | −3.56–17.56 | 0 |
| 15 | Sausages and ham | 6.07 | 6.45 | 2.84 | 0 | 10 | 3.04 | −1.15–14.05 | 0 |
| 16 | Milk drinks | 5.97 | 6.00 | 2.89 | 0 | 10 | 3.26 | −2.15–14.15 | 0 |
| 17 | Sour products | 5.88 | 6.00 | 3.00 | 0 | 10 | 3.85 | −3.64–15.64 | 0 |
| 18 | Beef and pork | 5.75 | 6.00 | 3.06 | 0 | 10 | 3.48 | −2.71–14.71 | 0 |
| 19 | Crisps | 5.65 | 6.25 | 3.23 | 0 | 10 | 3.85 | −3.39–15.89 | 0 |
| 20 | Fast food | 5.46 | 5.90 | 3.40 | 0 | 10 | 4.30 | −4.85–16.65 | 0 |
| 21 | Spicy dishes | 5.12 | 5.40 | 3.26 | 0 | 10 | 4.30 | −5.35–16.15 | 0 |
| 22 | Salty products | 4.98 | 4.80 | 3.03 | 0 | 10 | 4.00 | −5.21–14.81 | 0 |
| 23 | Milk soups[**] | 3.30 | 2.55 | 3.13 | 0 | 10 | 3.34 | −5.79–10.89 | 0 |
| 24 | Seafood | 2.99 | 1.50 | 3.28 | 0 | 10 | 2.22 | −4.06–7.06 | 31 |

**Notes.**
[*]This was broth, eaten the Polish way, with pasta.
[**]This is a sweet dish made by pouring hot milk over things such as: boiled rice, pasta, oatmeal, chocolate chips or corn flakes, *etc.*
Med, Median; SD, Standard Deviation; Min, Minimum; Max, Maximum; MAD, Median absolute deviation; RNO, Range of results which are not outliers; NRem, Number of cases removed as outliers.

# DISCUSSION

## Food types ranking

This research showed that women derived the greatest pleasure from eating fruits, sweet desserts, vegetables and salads, chocolate, and poultry (Table 2). These five most preferred food types are either products that are considered healthy (fruits, vegetables and salads, poultry) (*Mason-D'Croz et al., 2019*), or those that stimulate the reward system in the brain due to their high content of simple sugars (sweet desserts and chocolate) (*Kelley, 2004*; *Kelley, Schiltz & Landry, 2005*). The integration of multiple senses, including taste, smell, texture, temperature, sight and touch, that occurs in response to consumption of food, result in a taste perception experienced at multiple levels of the central nervous system (*Lorenzo, 2021*). Several areas of the brain are involved in this reward system, especially the mesolimbic system, reaching from the ventral tegmental area of the midbrain to the

**Table 3 Regression models with significant predictors.** Regression models in which at least one predictor variable (age, BMI, pack-years, olfactory sensitivity (dilution step), ability to name scents, hormonal status: C, contraception; F, follicular phase; O, ovulatory phase; L, luteal phase) affected the declared pleasure derived from eating a certain type of food.

| Dish (N) | $R^2_c$ | Predictor variables | B | PU | | t/F | eta$^2$ | p |
|---|---|---|---|---|---|---|---|---|
| Candies and jellybeans (128) | 0.15 | Age | −.40 | −.90 | .10 | −1.57 | .02 | .120 |
| | | BMI | −.27 | −.51 | −.03 | −2.24 | .04 | .027 |
| | | Pack-years | −.14 | −.90 | .62 | −.36 | <.01 | .717 |
| | | Olfactory sensitivity (dilution step) | .07 | −.13 | .28 | .72 | <.01 | .476 |
| | | Ability to name scents | .27 | −.29 | .83 | .94 | .01 | .349 |
| | | Hormonal status *** (C/F/O/L) | | | | 3.33 | .08 | .022 |
| Broth* (129) | 0.13 | Age | .35 | −.19 | .88 | 1.29 | .01 | .199 |
| | | BMI | −.34 | −.60 | −.08 | −2.61 | .05 | .010 |
| | | Pack-years | .08 | −.74 | .90 | .19 | <.01 | .848 |
| | | Olfactory sensitivity (dilution step) | .01 | −.20 | .23 | .13 | <.01 | .893 |
| | | Ability to name scents | .20 | −.41 | .80 | .64 | <.01 | .522 |
| | | Hormonal status *** (C/F/O/L) | | | | 2.97 | .07 | .035 |
| Sausages and ham (129) | 0.12 | Age | .01 | −.44 | .46 | .05 | <.01 | .958 |
| | | BMI | .01 | −.20 | .23 | .13 | <.01 | .894 |
| | | Pack-years | .08 | −.61 | .77 | .23 | <.01 | .815 |
| | | Olfactory sensitivity (dilution step) | .12 | −.06 | .31 | 1.33 | .01 | .187 |
| | | Ability to name scents | .79 | .28 | 1.30 | 3.09 | .07 | .013 |
| | | Hormonal status *** (C/F/O/L) | | | | 4.66 | .10 | .014 |
| Pasta (129) | 0.10 | Age | −.01 | −.40 | .38 | −.06 | <.01 | .954 |
| | | BMI | .04 | −.15 | .23 | .42 | <.01 | .679 |
| | | Pack-years | −.10 | −.70 | .51 | −.32 | <.01 | .753 |
| | | Olfactory sensitivity (dilution step) | −.07 | −.23 | .09 | −.90 | .01 | .369 |
| | | Ability to name scents | −.07 | −.52 | .37 | −.33 | <.01 | .743 |
| | | Hormonal status *** (C/F/O/L) | | | | 3.76 | .09 | .013 |
| Beef and Pork (129) | 0.08 | Age | .08 | −.43 | .60 | .32 | <.01 | .752 |
| | | BMI | <.01 | −.25 | .25 | .03 | <.01 | .980 |
| | | Pack-years | .25 | −.54 | 1.05 | .63 | <.01 | .528 |
| | | Olfactory sensitivity (dilution step) | −.06 | −.27 | .15 | −.57 | <.01 | .572 |
| | | Ability to name scents | .59 | .01 | 1.18 | 2.01 | .03 | .046 |
| | | Hormonal status *** (C/F/O/L) | | | | 1.76 | .04 | .159 |

**Table 3** (*continued*)

| Dish (N) | $R^2_c$ | Predictor variables | B | PU | | t/F | eta$^2$ | p |
|---|---|---|---|---|---|---|---|---|
| | | Age | .16 | −.34 | .67 | .64 | <.01 | .525 |
| | | BMI | .16 | −.34 | .67 | .64 | <.01 | .525 |
| Milk soups** (129) | 0.06 | Pack-years | .03 | −.22 | .27 | .21 | <.01 | .834 |
| | | Olfactory sensitivity (dilution step) | −.82 | −1.60 | −.04 | −2.09 | .03 | .039 |
| | | Ability to name scents | .14 | −.07 | .34 | 1.29 | .01 | .200 |
| | | Hormonal status *** (C/F/O/L) | .22 | −.36 | .79 | .75 | <.01 | .452 |

**Notes.**

Broth*, this was broth, eaten the Polish way, with pasta; Milk soup**, this is a sweet dish made by pouring hot milk over things such as: boiled rice, pasta, oatmeal, chocolate chips or corn flakes etc.; Hormonal status ***, after the removal of age outliers, there was no woman with menopause; N, number of samples without outliers; B, Unstandardized regression coefficients; PU, confidence interval; $R^2_c$, multiple determination coefficient; eta$^2$-effect size; t/F, t or F Statistic; p, level of significance.

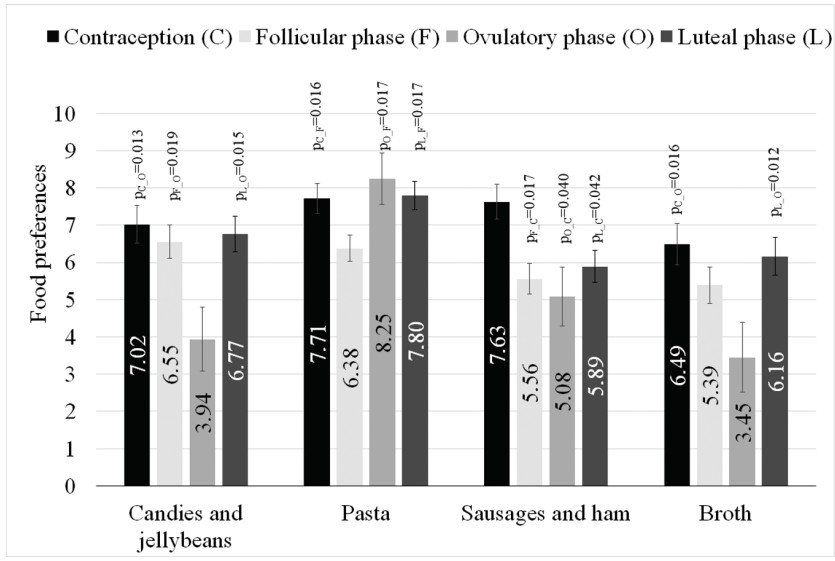

**Figure 1** **Estimated marginal means and standard errors of preference for types of food in women with different hormonal statuses, along with the intergroup differences.** Intergroup differences: *ps* from multiple comparisons with Sidak correction.

nucleus accumbens in the striatum and hippocampus. Many studies showed an increased brain stimulation in regions of the reward system following the exposure to a food stimuli that was highly palatable, when compared to less palatable one. In the absence of hunger, the activation of the mesolimbic reward system following consumption of palatable food reflects the hedonic appeal of those foods (*Kelley, 2004*; *Kelley, Schiltz & Landry, 2005*). Many studies indicate that the activation of the reward system is strong enough to promote food consumption beyond the physiological saturation of the body, which can lead to obesity (*Berthoud, 2004*; *Tordoff, 2002*; *Petrovich & Gallagher, 2007*).

A positive phenomenon noted in our study and important for women's health is the low ranking of unhealthy products such as crisps, fast food and salty products, which had 19th, 20th and 22nd place in the food preferences ranking, respectively. Food consumed out of home, such as fast food, or even at home, but ordered as a takeaway, is usually less healthy
**Table 4 Regression models with no significant predictors.** Regression models in which none of the predictor variables (age, BMI, pack-years, olfactory sensitivity (dilution step), ability to name scents, hormonal status: C, contraception, F, follicular phase, O, ovulatory phase, L, luteal phase) had a statistically significant impact on the declared pleasure derived from eating a certain type of food.

| Dish (N) | $R^2_c$ | Predictor variables | B | PU | | t/F | eta$^2$ | p |
|---|---|---|---|---|---|---|---|---|
| Sweet desserts (96) | 0.04 | Age | .03 | −.10 | .16 | .51 | <.01 | .614 |
| | | BMI | <.01 | −.07 | .07 | .06 | <.01 | .949 |
| | | Pack-years | .05 | −.21 | .31 | .37 | <.01 | .709 |
| | | Olfactory sensitivity (dilution step) | −.01 | −.06 | .05 | −.24 | <.01 | .813 |
| | | Ability to name scents | .08 | −.08 | .23 | 1.00 | .01 | .322 |
| | | Hormonal status* (C/F/O/L) | | | | .54 | .02 | .657 |
| Vegetables and salads (117) | 0.06 | Age | .20 | −.08 | .48 | 1.42 | .02 | .158 |
| | | BMI | .03 | −.11 | .17 | .48 | <.01 | .633 |
| | | Pack-years | .15 | −.31 | .61 | .66 | <.01 | .509 |
| | | Olfactory sensitivity (dilution step) | −.01 | −.13 | .10 | −.24 | <.01 | .809 |
| | | Ability to name scents | .05 | −.27 | .36 | .30 | <.01 | .767 |
| | | Hormonal status* (C/F/O/L) | | | | 1.03 | .03 | .383 |
| Chocolate (108) | 0.04 | Age | −.08 | −.31 | .16 | −.64 | <.01 | .521 |
| | | BMI | .05 | −.07 | .16 | .78 | .01 | .437 |
| | | Pack-years | −.03 | −.54 | .48 | −.11 | <.01 | .909 |
| | | Olfactory sensitivity (dilution step) | .03 | −.07 | .13 | .55 | <.01 | .584 |
| | | Ability to name scents | .07 | −.21 | .35 | .52 | <.01 | .606 |
| | | Hormonal status* (C/F/O/L) | | | | 1.06 | .03 | .368 |
| Poultry (124) | 0.08 | Age | .10 | −.25 | .45 | .58 | <.01 | .561 |
| | | BMI | .01 | −.16 | .17 | .09 | <.01 | .925 |
| | | Pack-years | −.05 | −.58 | .48 | −.19 | <.01 | .848 |
| | | Olfactory sensitivity (dilution step) | .12 | −.03 | .26 | 1.59 | .02 | .114 |
| | | Ability to name scents | .33 | −.08 | .74 | 1.59 | .02 | .115 |
| | | Hormonal status* (C/F/O/L) | | | | 1.98 | .05 | .121 |
| Sour products (129) | 0.10 | Age | −.06 | −.55 | .44 | −.22 | <.01 | .823 |
| | | BMI | −.06 | −.30 | .17 | −.53 | <.01 | .599 |
| | | Pack-years | −.45 | −1.21 | .31 | −1.17 | .01 | .243 |
| | | Olfactory sensitivity (dilution step) | −.20 | −.40 | <.01 | −1.96 | .03 | .052 |
| | | Ability to name scents | .40 | −.16 | .96 | 1.40 | .02 | .164 |
| | | Hormonal status* (C/F/O/L) | | | | 1.77 | .04 | .157 |
| Dumplings (129) | 0.04 | Age | −.12 | −.56 | .31 | −.56 | <.01 | .576 |
| | | BMI | −.02 | −.23 | .19 | −.18 | <.01 | .857 |
| | | Pack-years | −.03 | −.70 | .63 | −.10 | <.01 | .921 |
| | | Olfactory sensitivity (dilution step) | .06 | −.12 | .24 | .65 | <.01 | .516 |
| | | Ability to name scents | .10 | −.39 | .59 | .40 | <.01 | .689 |
| | | Hormonal status* (C/F/O/L) | | | | 1.45 | .04 | .231 |

**Table 4** (*continued*)

| Dish (N) | $R^2_c$ | Predictor variables | B | PU | | t/F | eta² | p |
|---|---|---|---|---|---|---|---|---|
| Egg dishes (127) | 0.04 | Age | −.08 | −.49 | .34 | −.37 | <.01 | .710 |
| | | BMI | .13 | −.07 | .34 | 1.31 | .01 | .192 |
| | | Pack-years | −.43 | −1.08 | .23 | −1.29 | .01 | .199 |
| | | Olfactory sensitivity (dilution step) | −.04 | −.22 | .13 | −.52 | <.01 | .604 |
| | | Ability to name scents | .02 | −.47 | .52 | .09 | <.01 | .925 |
| | | Hormonal status* (C/F/O/L) | | | | .77 | .02 | .513 |
| Cheese (129) | 0.06 | Age | .24 | −.25 | .73 | .98 | .01 | .330 |
| | | BMI | .04 | −.20 | .27 | .33 | <.01 | .741 |
| | | Pack-years | .11 | −.64 | .86 | .29 | <.01 | .770 |
| | | Olfactory sensitivity (dilution step) | <.01 | −.20 | .20 | .01 | <.01 | .995 |
| | | Ability to name scents | .09 | −.46 | .64 | .33 | <.01 | .745 |
| | | Hormonal status* (C/F/O/L) | | | | 2.08 | .05 | .106 |
| Soups (129) | 0.06 | Age | .23 | −.23 | .70 | .99 | .01 | .324 |
| | | BMI | −.05 | −.28 | .17 | −.46 | <.01 | .649 |
| | | Pack-years | −.17 | −.89 | .55 | −.47 | <.01 | .638 |
| | | Olfactory sensitivity (dilution step) | −.08 | −.27 | .11 | −.87 | .01 | .386 |
| | | Ability to name scents | .46 | −.07 | .98 | 1.71 | .02 | .090 |
| | | Hormonal status* (C/F/O/L) | | | | .67 | .02 | .572 |
| Milk drinks (128) | 0.09 | Age | .05 | −.19 | .30 | .41 | <.01 | .685 |
| | | BMI | −.02 | −.79 | .76 | −.05 | <.01 | .963 |
| | | Pack-years | .07 | −.14 | .28 | .67 | <.01 | .505 |
| | | Olfactory sensitivity (dilution step) | .16 | −.41 | .73 | .57 | <.01 | .571 |
| | | Ability to name scents | | | | 1.17 | .03 | .324 |
| | | Hormonal status* (C/F/O/L) | .05 | −.19 | .30 | .41 | <.01 | .685 |
| Fruits 92 | 0.07 | Age | .04 | −.07 | .16 | .77 | .01 | .444 |
| | | BMI | −.04 | −.10 | .01 | −1.57 | .03 | .121 |
| | | Pack-years | .01 | −.15 | .16 | .11 | <.01 | .916 |
| | | Olfactory sensitivity (dilution step) | .01 | −.04 | .05 | .32 | <.01 | .747 |
| | | Ability to name scents | .08 | −.03 | .19 | 1.39 | .02 | .169 |
| | | Hormonal status* (C/F/O/L) | | | | .17 | .01 | .916 |
| Fast food (129) | 0.06 | Age | −.21 | −.72 | .31 | −.79 | .01 | .433 |
| | | BMI | .07 | −.18 | .32 | .58 | <.01 | .564 |
| | | Pack-years | .63 | −.17 | 1.43 | 1.56 | .02 | .120 |
| | | Olfactory sensitivity (dilution step) | −.05 | −.27 | .16 | −.50 | <.01 | .618 |
| | | Ability to name scents | .28 | −.31 | .86 | .93 | .01 | .353 |
| | | Hormonal status* (C/F/O/L) | | | | .99 | .02 | .402 |

**Table 4** (*continued*)

| Dish (N) | $R^2_c$ | Predictor variables | B | PU | | t/F | eta$^2$ | p |
|---|---|---|---|---|---|---|---|---|
| Spicy dishes (129) | 0.05 | Age | .27 | −.28 | .81 | .97 | .01 | .333 |
| | | BMI | .14 | −.12 | .41 | 1.07 | .01 | .288 |
| | | Pack-years | −.06 | −.90 | .79 | −.13 | <.01 | .897 |
| | | Olfactory sensitivity (dilution step) | −.03 | −.25 | .19 | −.27 | <.01 | .785 |
| | | Ability to name scents | .10 | −.52 | .72 | .33 | <.01 | .741 |
| | | Hormonal status* (C/F/O/L) | | | | 1.21 | .03 | .309 |
| Salty products (129) | 0.02 | Age | −.06 | −.55 | .44 | −.22 | <.01 | .824 |
| | | BMI | −.05 | −.29 | .19 | −.41 | <.01 | .682 |
| | | Pack-years | .01 | −.75 | .76 | .02 | <.01 | .988 |
| | | Olfactory sensitivity (dilution step) | −.06 | −.26 | .15 | −.55 | <.01 | .581 |
| | | Ability to name scents | .30 | −.26 | .86 | 1.07 | .01 | .285 |
| | | Hormonal status* (C/F/O/L) | | | | .32 | .01 | .813 |
| Seafood (107) | 0.05 | Age | .20 | −.25 | .64 | .88 | .01 | .380 |
| | | BMI | .08 | −.09 | .26 | .94 | .01 | .350 |
| | | Pack-years | .20 | −.37 | .77 | .69 | .00 | .494 |
| | | Olfactory sensitivity (dilution step) | .08 | −.08 | .24 | 1.02 | .01 | .311 |
| | | Ability to name scents | .22 | −.21 | .66 | 1.03 | .01 | .308 |
| | | Hormonal status* (C/F/O/L) | | | | .08 | <.01 | .971 |
| Bread (128) | 0.06 | Age | .11 | −.25 | .47 | .63 | <.01 | .531 |
| | | BMI | −.06 | −.23 | .11 | −.70 | <.01 | .488 |
| | | Pack-years | .04 | −.50 | .58 | .15 | <.01 | .879 |
| | | Olfactory sensitivity (dilution step) | .08 | −.07 | .22 | 1.08 | .01 | .283 |
| | | Ability to name scents | .14 | −.25 | .54 | .72 | <.01 | .475 |
| | | Hormonal status* (C/F/O/L) | | | | 2.22 | .05 | .089 |
| Fish dishes 129 | 0.07 | Age | .30 | −.15 | .75 | 1.32 | .01 | .191 |
| | | BMI | −.03 | −.25 | .19 | −.28 | <.01 | .779 |
| | | Pack-years | −.45 | −1.15 | .24 | −1.29 | .01 | .200 |
| | | Olfactory sensitivity (dilution step) | .07 | −.12 | .25 | .75 | <.01 | .456 |
| | | Ability to name scents | .30 | −.21 | .81 | 1.17 | .01 | .245 |
| | | Hormonal status* (C/F/O/L) | | | | .77 | .02 | .510 |
| Crisps (129) | 0.07 | Age | −.25 | −.73 | .23 | −1.03 | .01 | .304 |
| | | BMI | −.09 | −.33 | .14 | −.81 | .01 | .419 |
| | | Pack-years | .20 | −.54 | .93 | .53 | <.01 | .595 |
| | | Olfactory sensitivity (dilution step) | −.02 | −.21 | .18 | −.16 | <.01 | .876 |
| | | Ability to name scents | .04 | −.50 | .58 | .14 | <.01 | .887 |
| | | Hormonal status* (C/F/O/L) | | | | 2.05 | .05 | .111 |

**Notes.**
An asterisk (*) indicates hormonal status, after the removal of age outliers, there was no woman with menopause; N, number of samples without outliers; B, Unstandardized regression coefficients; PU, confidence interval; $R^2_c$, multiple determination coefficient; eta$^2$, effect size; t/F, t or F Statistic; p, level of significance.

than meals prepared at home (*Lachat et al., 2012*). Such foods not only tend to be high in saturated fat and salt and of low nutritional value (*Burgoine et al., 2016*; *Drewnowski & Specter, 2004*), but are also served in large portions (*Young & Nestle, 2002*). Research by *Frensh, Harnack & Jeffery (2000)* found that regular fast food consumption was associated with weight gain in adults.

## Sense of smell

The analysis of the olfactory efficiency impact on food preferences in women showed that out of 24 food types, significant correlations were found only for two of them (Table 3). Women with a better sense of smell (manifested as their ability to name odors) declared greater pleasure derived from consuming sausages and ham, beef and pork. It was not possible to confirm the correlation between preferences and the perception of smell sensations using both olfactory tests simultaneously for any of the specified food types. According to the literature data, such weak dependencies seem surprising. The importance of the sense of smell in eating behavior has been emphasized by numerous researchers, most often with regard to age-related chemosensory changes (*Gopinath et al., 2016*). Smell plays an important role in the perception of food, and olfactory or taste dysfunctions can therefore influence eating behaviors. This, in turn, directly influences the quality of everyday life. Furthermore, olfactory dysfunction is associated with a reduced ability to taste food, as physiologically there is a spatial overlap in taste and olfactory perception (*Zang et al., 2019*). It was noticed that people suffering from anosmia used more sugar, ketchup, mayonnaise, or sour cream, which could function as a compensatory mechanism. Thus, it was confirmed that a decrease in sensitivity to taste in response to olfactory dysfunction indicates a high interaction between olfaction and taste. In the present study, the median age of the women was only 23 years, the olfactory detection threshold coincided with the population average, and the spread of the olfactory sensitivity results was relatively small. So, presumably, the preferences of only two out of 24 assessed food types depended on the olfactory efficiency of the studied women.

## Age

The use of regression models in the assessment of factors affecting food preferences showed that the declared pleasure derived from eating different types of food in the group of women not depended on their age (Table 3). These results are apparently contrary to results obtained by other authors, who indicate numerous biological factors that may lead to changes in food preferences occurring with age. In their study, *Guido, Perna & Carrai (2016)*, found that the increasing age is associated with decreasing responsiveness to NaCl and sweet solutions, and a reduced ability to perceive smells. *Barragan et al. (2018)* conducted a study in a healthy population of people aged 18 to 80 years, and demonstrated a significant decrease in the perception of different tastes with age. Moreover, the decrease in the intensity of taste perception was significant for all of them. With regard to food preferences, *Guido, Perna & Carrai (2016)*, found that the elderly like ''vegetables'' and ''fruits'' but dislike ''spicy'' more than their younger counterparts.

It is worth emphasizing that in the research presented in this study, due to elimination of outliers, the observations were conducted solely in young women in the age range (RNO)
of 19.29–26.71 years, and in this age range, no influence of the age on food preferences was observed. When the analysis was conducted without outliers elimination, in the age group of 18–75 years, it was observed that the declared pleasure derived from eating crisps, milk soups and fruit depended on women's age (Supplement 3).

## BMI

The food preferences of women with a high BMI value turned out to be surprising. Compared to slimmer women, these women declared lower pleasure derived from eating candies and jellybeans and, furthermore, they had lower preference for broth (Table 3). Considering that broth (served the Polish way, with noodles) and, above all, candies are high-calorie foods (Hall, 2016), it could be assumed that overweight and obese people would rather prefer those food types. Such surprising preferences may be related to the fact that overweight and obese women often try to lose weight. Perhaps these repeated attempts to avoid calorie-rich foods have resulted in partial success and decreased the pleasure derived from eating candies and jellybeans and broth. It cannot be excluded that overweight and obese women wanted to present themselves in a better light and therefore, indicated preference for those food types at a level lower than their actual preferences.

In the paper of Guido, Perna & Carrai (2016), the BMI influence on preferences for various Italian food types was analyzed. No statistically significant relationships were found between the preferences and BMI. Only a negative correlation at the level of $p = 0.075$ was observed between BMI and "fruit foods", which included: fruit, cake, citrus fruit, and a positive relationship at the level of $p = 0.084$ was observed between BMI and dairy foods, which included: Butter, Cream, Fresh cheese and Hard cheese. Comparing these observations with our studies, it is possible to confirm a certain convergence in the direction of the observed relations. Sweet foods and fruits were negatively associated with BMI, and cheese showed a positive relationship with BMI. However, it should be strongly emphasized that these results did not reach statistical significance. The lack of clear relations between food preferences and BMI indicates that the formed food preferences do not translate directly into what we actually eat.

Food consumption is regulated by a complex physiological interaction and by environmental and cognitive factors. In most developed countries, it is easy to obtain enough food to maintain caloric homeostasis through the wide availability of inexpensive and palatable foods, but, unfortunately, the most pleasurable foods are often high in calories. The sensory properties of food and drink are experienced before, during and after consumption (McCrickerd & Forde, 2016). Many food establishments also use tempting aromas to attract potential customers (Chebat & Michon, 2003). Laboratory evidence has shown that very pleasant food aromas can stimulate salivation and promote appetite and future consumption (Engelen et al., 2003). While looking at the causes of obesity, it should also be noted that there is a certain kind of conflict between retailers and consumers, and thus between obesity prevention and commercial policies. A popular "kebab or hamburger stand" is a source of income for someone and perhaps a livelihood of a whole family. Therefore, this type of food will always be available and will always attract with its smell and taste.

## Sex hormone levels

In women, sex hormone levels are constantly changing throughout their lives. These changes are most apparent in adolescence and during menopause, but fluctuations are also present during the menstrual cycle (*Farage, Osborn & MacLean, 2008*). The level of sex hormones is also modified through the use of hormonal contraception. Hormonal changes cause a number of physiological changes. For example, the use of hormonal contraception results in a noticeable increase in body weight (*Bonny, Secic & Cromer, 2011*). Perhaps the changes in body weight after the introduction of contraception are in some way related to the food preferences presented in this study. The declared pleasure derived from eating sausages and ham candies and jellybeans, pasta and broth differed significantly, depending on the women's hormonal status (Table 3). The greatest pleasure from eating sausages and ham was declared by women using hormonal contraception, and it was significantly higher when compared to the preferences of women at the follicular, ovulatory and luteal phases. In women using hormonal contraception, the declared pleasure from eating was higher when compared to preferences of women in the ovulatory phase also for candies and jellybeans, of women in the follicular phase for pasta, and of women in the ovulatory phase for broth.

In general, in the studies the highest declared pleasure from eating sausages and ham was demonstrated in women using the contraception, when compared to women in all phases of their monthly cycle. The lowest declared pleasure from eating pasta was seen in women in the follicular phase when compared to women in the ovulatory phase, luteal phase and those using hormonal contraception. Furthermore, the lowest declared pleasure from eating candies and jellybeans was seen in women in the ovulatory phase when compared to women in the follicular phase, luteal phase and those using hormonal contraception, and the lowest declared pleasure from eating broth was seen in women in the ovulatory phase when compared to women in the luteal phase and those using hormonal contraception.

The fluctuations in sex hormones have a role in various brain functions, such as sensory processing, and appetite (*Farage, Osborn & MacLean, 2008*). Previous studies have shown that taste perception, food preferences, and food cravings in women are not uniform across the menstrual cycle (*Gorczyca et al., 2016*). Food preferences and, therefore, the sense of taste and smell, are influenced by the phases of the menstrual cycle (*Stanić et al., 2021*). However, available studies are inconclusive. Some (*Navarrete-Palacios et al., 2003*) indicate that the olfactory recognition threshold changes and increases in sensitivity during the ovulation or luteal phase and decreases in sensitivity during the menstruation or follicular phase, but other studies have presented opposing results. Certain studies (*Derntl et al., 2013*) showed that olfactory sensitivity in the luteal phase was lower when compared to the follicular phase. In addition, it was demonstrated that the olfactory sensitivity in women using hormonal contraception was comparable to the olfactory sensitivity in women in the luteal phase of the menstrual cycle. In one study (*Stanić et al., 2021*). women who used oral contraceptives rated higher the sweet taste of sucrose. There is also a noticeable peak during the monthly cycle, with sweetness scoring higher in the hedonic assessment in the mid-luteal phase. The changes in olfactory sensitivity during the menstrual cycle are of a

linear nature. The women's olfactory sensitivity is the highest at the end of the cycle and the lowest in the follicular phase.

Studies (*Kammoun et al., 2017*) have shown that during the late luteal phase and the early follicular phase, women gained a significant amount of weight when compared to the peri-ovulatory phase. During the peri-ovulatory and luteal phases, the total calorie intake was notably higher than in the follicular phase. Dietary surveys showed a significant increase in fat, protein and carbohydrate intake during the peri-ovulatory and late luteal phases, when compared to the follicular period. Studies (*Hormes & Timko, 2011*) have shown significant differences in energy consumption and fluctuations in the appetite in women at different phases of the menstrual cycle. These phenomena are explained by the effects of estrogen and progesterone on gastric emptying and the secretion of gastrointestinal hormones, such as glucagonlike-peptide-1 (GLP-1) and cholecystokinin (CCK), which have a significant effect on appetite regulation. Studies (*Bryant, Truesdale & Dye, 2006*) showed that women consume about 100 kcal/day more in the premenstrual period, when compared to the rest of the menstrual cycle. Other studies (*Hormes & Rozin, 2009*) also showed that the preference for the taste of chocolate is particularly related to the luteal phase in the menstrual cycle. The research by *Klump et al. (2013)*, evaluated associations between internal changes in ovarian hormones levels and emotional eating across the menstrual cycle. The relation with estrogen and progesterone levels remained significant in the presence of other explanatory variables, which indicated the significance of sex hormones in so-called emotional eating during the menstrual cycle. Even when other explanatory variables existed, the fact that the interactions between estrogen and progesterone remained significant, serves to highlight how important these ovarian hormones are in the regulation of emotional eating throughout the menstrual cycle. The mid-luteal phase is the peak time for emotional eating and may be predicted by variations in the estrogen and progesterone levels.

## Smoking

This research showed that smokers declared they derived less pleasure from eating milk soups than non-smokers (Table 3). It is difficult to explain why lower declared pleasure from eating milk soups is associated with an increase in the habit intensity, but it is known that smoking negatively influences the sense of taste and the gastrointestinal tract function (*Audrain-McGovern & Benowitz, 2011*).

Furthermore, *Audrain-McGovern & Benowitz (2011)*, showed that smokers generally weigh 4–5 kilograms less than non-smokers, and that smoking reduces appetite and increases satiety. Interestingly, studies by *Palaniappan et al. (2001)*, showed that smokers consume more fat and calories than non-smokers. *Cooper et al. (2006)* demonstrated that smokers are aware of the relationship between smoking and body weight. 50% of female smokers and 26% of male smokers are afraid of weight gain after quitting, and these concerns are justified, as 49% of smokers gain weight after quitting smoking.

The relationship between cigarette smoking and a tendency for unhealthy eating behaviors is not clear. Nicotine increases the need for alcohol consumption, while a cross-substance craving as well as a desire for highly palatable foods has been demonstrated

for smokers (*Pepino & Mennella, 2014*). Studies (*Pepino & Mennella, 2014*) found that obese smokers have an altered perception of the taste of fat, which may result in an altered perception of fat-rich foods. Studies (*Chao et al., 2017*) also demonstrated that smokers more often report a need to eat, especially fat-rich foods. Smokers have also reported a habitual consumption of fat-rich foods and junk food. Research has confirmed previous findings that smokers not only have a more frequent craving for fat-rich foods, but also need them in larger quantities. However, the relationship between smoking and overall appetite turned out to be non-significant after adjustments for stress and depressive symptoms in the above-mentioned studies. As shown in neurobiological models of addiction, it is possible that nicotine activates the stress and reward areas that trigger feedback processes. Smoking and nicotine use induces the release of stress hormones, including corticotropin-releasing factor and glucocorticoids, and these hormones also promote the consumption of palatable and calorie-rich foods (*Rohleder & Kirschbaum, 2006*). Research results assessing the influence of smoking on the perception of smell and taste remain contradictory. However, the majority of them implies that cigarette smoking may have an adverse effect on the perception of smell. Some studies have failed to identify a significant effect of smoking on the impairment of smell or taste (*Liu et al., 2016*; *Guido, Perna & Carrai, 2016*). Other research (*Ajmani et al., 2017*) indicated that active smokers are at an increased risk of olfactory dysfunction, but this not concerns ex-smokers, and the smokers also have an impaired perception of sweet, salty, sour and bitter taste.

## Limitation

This publication describes the food preferences of women, taking into account 24 food types. Using a multivariate analysis, an attempt was made to determine how factors such as age, BMI, smoking frequency (pack-years), the sense of smell effectiveness (olfactory recognition threshold and smell identification test), and the hormonal status (menopause, hormonal contraception, phase of the menstrual cycle), when considered jointly and separately, explain the variability of the pleasure derived from eating a specific food type. Regression models were built and statistically significant factors were found which influenced the preference for certain types of food. However, it should be remembered that not only the statistical significance of the result (*p*-value) is important. Using the multiple determination coefficient (R2), it was possible to establish that in the best explained regression model (food type: candies and jellybeans), all examined factors together explain only 15% of the variance volatility. In this model, BMI (on the basis of eta-squared) was decisive for only 4% variance volatility and hormonal status for 8% variance volatility. This means that this research only touched the "tip of the iceberg", and the number and the importance of other factors influencing food preferences, which have been not described here, are, indeed, very large.

Self-reported food preferences may be burdened with the error of "social approval" resulting from social desirability bias, *i.e.,* a tendency of respondents to provide answers that are most socially acceptable, and will be perceived as appropriate, attractive, and in the case of food, "healthy". Respondents answering in a way that "controls social attractiveness", at their discretion, demonstrate a relevant nutritional knowledge, avoid

possible criticism, base their responses on the general knowledge they have about a typical healthy diet, and consider such responses to be socially desirable and even expected. Therefore, prejudices related to the social desire of certain eating behaviors may influence the results of research and should therefore be taken into account. Education in and awareness of proper eating patterns is one of the factors that may influence the obtained results (*Kowalkowska & Poínhos, 2021*). For example, *Devonport, Nicholls & Fullerton (2019)* demonstrated that women who volunteered for the studies scored lower than the control in terms of uncontrolled eating and emotional eating, which are eating behaviors commonly known to be socially undesirable. The study by *Miller et al. (2008)* found that self-reported preferences for consumption of fruit and vegetables are susceptible to significant prejudices related to social approval. It has been hypothesized that a group of women who believed that the purpose of the study was to measure the consumption of (healthy) fruit and vegetables might report higher fruit and vegetable consumption than those in the control group who believed the study had a more general purpose. Moreover, the study by *Horner et al. (2002)* showed that undergraduate students who received a brief nutrition message about fat and health showed significantly lower fat consumption in a survey than students who had not received such message before. The research presented in this paper, to some extent, may be burdened with measurement errors described above, for example, because a significant group of study participants are students of medical faculties who know the basics of healthy eating behaviors. Efforts were made to reduce the influence of the factors described above in this study. It was emphasized that participants should indicate how much they liked a certain type of food, not how often they ate them. It was also emphasized that responses should not be guided by the opinion whether it is allowed to eat such types of food and whether they are healthy. An additional factor that reduced the risk of social influence was the fact that the participants themselves viewed the photo album of the dishes and completed the answer sheets in writing; they did not have to answer the researcher, for example, that they liked desserts at 10, so the pressure of the researcher to immediately assess women's food preferences was reduced. During the study of food preferences, for each of the 24 food types, several products were presented and participants were asked to choose the tastiest, which additionally emphasized the hedonic aspect of the presented food type. Despite the countermeasures, the impact of public acceptance may have been present. The conclusions presented below should be interpreted only in context of limitations.

## CONCLUSIONS

Among women in Poland, the top five most liked food types are fruits, sweet desserts, vegetables/salads, chocolate and poultry. To confirm the extent to which the declared pleasure derived from eating these food types translates into health condition, further research on the consumption of these food types is necessary. In six out of twenty-four food types the impact of the sense of smell, BMI, smoking, or the hormonal status (phase of the menstrual cycle, hormonal contraception) on the declared pleasure derived from eating was observed. The hormonal status was the factor with the greatest influence on

food preferences. In the light of the presented food preferences of women in Poland and the influence of individual factors on food preferences, it is not possible to unequivocally identify the elements which predispose women to a greater risk of developing diseases of civilization due to their food preferences. Further more detailed research is needed to confirm the real influence of different factors on the consumption of food types such as, sausages and ham, milk soups, candies and jellybeans, broth, beef and pork, and pasta.

### Funding

This work was supported by the Medical University of Silesia in Katowice. The funders had no role in study design, data collection and analysis, decision to publish, or preparation of the manuscript.

### Grant Disclosures

The following grant information was disclosed by the authors:
The Medical University of Silesia in Katowice.

### Competing Interests

The authors declare there are no competing interests.

### Author Contributions

- Magdalena Hartman-Petrycka conceived and designed the experiments, performed the experiments, analyzed the data, prepared figures and/or tables, authored or reviewed drafts of the article, and approved the final draft.
- Joanna Witkoś analyzed the data, authored or reviewed drafts of the article, and approved the final draft.
- Agata Lebiedowska performed the experiments, authored or reviewed drafts of the article, and approved the final draft.
- Barbara Błońska-Fajfrowska conceived and designed the experiments, performed the experiments, authored or reviewed drafts of the article, and approved the final draft.

### Human Ethics

The following information was supplied relating to ethical approvals (i.e., approving body and any reference numbers):
The research project was approved by the Bioethics Committee of the Medical University of Silesia (KNW/0022/KB1/47/12).

### Ethics

The following information was supplied relating to ethical approvals (i.e., approving body and any reference numbers):
The research project was approved by the Bioethics Committee of the Medical University of Silesia (KNW/0022/KB1/47/12).

## Data Availability

The raw measurements are available in the Supplementary Files.

## Supplemental Information

Supplemental information for this article can be found online at http://dx.doi.org/10.7717/peerj.13538#supplemental-information.

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
