# Peer review of "Individual characteristics, including olfactory efficiency, age, body mass index, smoking and the sex hormones status, and food preferences of women in Poland"

_PeerJ, doi:10.7717/peerj.13538_

## Round 0.1 · original submission · Major Revisions

I have received three reviews of your paper and they are in agreement that the methods are not fully described and that the conclusions are not warranted by the findings. The former issue--lack of detail in methods--might be a simple fix by providing more detail, but I am concerned that the issues raised may not be satisfied when those details are provided The latter issue--expansive and over-reaching conclusions--may be addressed by moving some of the material to the introduction and discussion and by tightening the focus but a new conclusion is needed that addresses the results of the specific study conducted.

I agree with the reviewers that there are interesting data here that can support the exploratory research question posed regarding factors affecting women's food preferences. I also agree that methods are lacking which may be why some of the linkages between findings and conclusions are extremely tenuous and/or not warranted by the data.

Each reviewer has specific concerns that should be addressed in addition to the primary concerns noted above. The revisions requested may require that a new paper be written.

·

Basic reporting

Dear authors, I have read the manuscript with great interest. I found this study setting out to test an interesting research question: whether certain factors affect liking of different foods. The research question is interesting in that it could help understand what drives healthy and unhealthy food choice. The reporting is clear until the discussion, although the English is sometimes somewhat colloquial and may be improved to improve the clarity of the manuscript. The data was shared with the article, and the data structure and variables are clear. However, I have a number of questions with respect to the methods, a number of concerns regarding the data-analysis, and I believe the discussion section would need to be largely re-written and shortened before it is suitable for publication in PeerJ.
I nevertheless hope that my (many) comments find you in good spirit, and will allow improvement of the manuscript:

For the abstract, my suggestion would be to summarize instead of reporting results with statistics.

Paragraph in lines 101-108 would benefit from additional literature sources and more explanation on the causal effects drawn.

I would probably not call ‘chocolate’, ‘fruit’, ‘candies and jellybeans’ a ‘dish’, but a food type or food item instead.

The discussion is relatively long with respect to the other sections of the manuscript: it covers half of the main body of text. The discussion is speculative at places, and touches upon many topics not directly relevant for the study at hand. The discussion does not cover the same content and literature as the introduction.

Experimental design

The experimental design is correlational, which matches the research question and analyses proposed. The research fills a gap of knowledge.
With respect to the methods, I do have a few questions, specifically:

Lines 125-134: how was the sample size determined? Was there an a-priori expectation for the effect sizes?

Lines 132-134 are not immediately clear: How can people with high olfactory sensitivity be classified as anosmic?

Lines 146-155. On what criteria were these food types selected for the study? In addition: Having access to the pictures of the food types would help understand the methods and results of the study.

Line 155-156: the anchor labels do not make a lot of sense to use in English – perhaps good to also report the original Polish scale anchors.

The description of the threshold test is not very clear:
Line 162, 168, etc: In my understanding, a fragrance would mean a (complex) mixture of odorous molecules. It seems ‘fragrance’ is used here synonymous to ‘smell’ or ‘odor’.
Line 165: N-butanol is used, but presumably at different concentrations and not at a fixed concentration of 59.9 PPM. Or is the concentration of 59.9 PPM used as the first step (step 0) in the dilution series?
Line 166: air is not an inert gas.
Line 167: the sentence ‘providing a correct, positive answer twice in a row by all participants in the study ended the measurement cycle’ is not clear. Usually, in this type of test, a staircase procedure is used, in which the concentrations alternate between higher and lower dilution steps, but is not clear whether this was also done here? It is not clear what the task of the participants was; what constitutes a trial?

Lines 174-179: was the task to match odors to a name (i.e., multiple choice), or to free name the odors? In case of the latter, how are correct answers determined (in other words, would ‘lemon’, ‘citrus’, ‘7-up’, ‘limonene’ or ‘lemon scented cleaning product’ all count as a correct answers to the limonene odor?)

Table 1 & 2: It is surprising that fruit, and vegetables and salads are in the top 3 most liked food types. Additionally, I find it very surprising that the participants derived more pleasure from eating vegetables and salads than chocolate. This could be a translation issue (was it really ‘derived pleasure’ that was asked from the participants?), or, more damning for the conclusions of the article, it may be due to some type of response bias, e.g., demand characteristics or social desirability. Could the authors comment on this?

Table 1: Something seems odd about the n-butanol dilution step mean and standard deviation in Table 1. It does not make sense to calculate the average of a logarithmic scale. This leaves me to wonder: on what scale was olfactory sensitivity added to the analyses?

Validity of the findings

Regarding the validity of the findings, I have a number of concerns. It is sometimes not entirely clear how the analyses were performed, and whether the analyses are fit for the types of data used.

Specific comments are:
Lines 186-202: what was the nominal variable in the regression analysis? What was the dependent variable in each regression model? What is ‘ability to associate smells’? It might be good to spell out the levels of each predictor variable (e.g., how can participants differ on hormonal status in context of this study?).

In light of the research question, it may be informative to group food types into a few categories (e.g., ‘healthy’ and ‘unhealthy’, snacks vs meals, or by general food group) of which the means may be used in a few regression models, instead of running regression analyses for each food type separately.

It is not entirely clear what kind of regression analyses were done. What do t and F values denote in this context?
Were assumptions of the regression models checked beforehand? For example, the variable age is not normally distributed, with many more younger participants than older participants.

How were outliers handled, and missing data? I might imagine the person who has 60 pack years on a group average of .8 (sd = 4.7) is removed from the analyses?
Many of the significant p-values are close to p = .05, with relatively small effects. Was there some control for multiple testing?

Many of the points in the discussion are not directly supported by the data. In particular:
Lines 247-248: this relationship is speculatory, and the causal direction cannot be inferred from the data (e.g., vegetarians may rate meats as less pleasurable because they are vegetarians, instead of eating vegetarian because they don’t like meat). Also, the comparison to men can not be made with the current dataset. Was data on diet preferences collected in the present study? This conclusion is not supported by the data.
Lines 263-266: the only significant effects of BMI were on candies and on broth, and in both cases this relationship was negative, meaning that higher BMI resulted in lower liking of candies and broth, respectively. This conclusion is not supported by the data.
Lines 272-275: this relationship is plausible, but given that there is no data on consumption frequency of each of the food items, not directly supported by the data.
Lines 267-287: the history of Polish cuisine is interesting, but seems not directly relevant for the article.
Lines 288-297: please comment on whether this finding may be due to social desirability in answering the questions.
Lines 296-316: sense of smell is measured in two ways in the study, using olfactory sensitivity and naming ability. Only one of these (naming ability) showed significant relationships with food liking, but odor sensitivity is generally taken as a more basic measure of sense of smell. The general conclusions in these paragraphs are therefore not fully supported by the data.

Additional comments

additional specific comments:
line 56-57: it is not immediately clear what the percentages in this sentence mean
Line 112: I would maybe change this sentence to ‘…some considered less healthy with high energy value and some considered healthy, such as vegetables, fruit and fish.”
Line 155: a visual analogue scale?

Use of language – can be improved. A few examples (not exhaustive):
Line 21: ‘are’ or ‘mostly depend’
Line 46: remove ‘healthy’ and ‘unhealthy’
Line 48: remove ‘only’
Line 55: suggestion to remove ‘nowadays’
Line 161: remove ‘the’
Line 200: ‘predictrs’
My suggestion would be to change ‘women’ to ‘participants’ in the description of table 1.

·

Basic reporting

The article must conform to professional standards of courtesy and expression, I would suggest you go through the report to check for incorrect expressions or technical terms.

The article should include sufficient introduction and background to demonstrate how the work fits into the broader field of knowledge and knowledge gap. Your introduction needs more detail - provide more justification for your study - hormonal levels and impact on smell which would potentially affect food preferences i.e expanding on the knowledge gap to be filled. Relevant prior literature should be appropriately referenced throughout the report.

All data has been made available in accordance with PeerJ Data Sharing policy.

Experimental design

The methods should be described with sufficient information to be reproducible by another investigator. It is not clear how the participants were recruited, what are the demographics of the group of women (transgender participants were they excluded or included), and how health interviews were conducted and data collected, and how anthropometric measurements taken and/ or data collected.

Validity of the findings

All underlying data have been provided; they are robust and statistically sound. Although the results are compelling, the data analysis could have consider factor analysis - principle component analysis and could have checked any correlations of some of the predictors which were not quite clear from your results. The discussion on correlation of predictors e.g sense of smell and greater pleasure should have been accompanied by results or reference to the tables with the results. The discussions should be focused on your results and backed up with literature and not vice versa. Conclusion are well stated and indicated were correlations could have occurred.

Additional comments

This is a very interesting research which would contribute to dissemination of science on impact of sense of smell on food preference and health. If they would need more detailed feedback on any of the sections, PeerJ should feel free to contact me.

Reviewer 3 ·

Basic reporting

I regard the manuscript to fullfil the criteria of basic reporting set by the PeerJ journal.

Experimental design

Previously published literature on food preferences is large. The authors should make it clearer what is the knowledge gap they aim to fill. Is it food preferences of Polish women? I guess food preferences in many cultures have already studied.

For validation of the results from the olfactory test, it is great that the participants' nasal patency and symmetry of air flow through the nostrils were confirmed. The methods to measure olfacoty sensitivity (as detection threshold to n-butanol, a standard odorant) and odor identification test (althought employing only five odors) seem valid.

In contrast, the authors merely state (lines 143-144) "anthropometric data -- were collected." Were values for height and weight measured by the researchers or requested to self-report by the participants?

Regarding the assessment of food preferences it is good that the authors have explained the food items that are local (Polish) specialties. The authors mention that the food categories (total 24) for which the pleasantness were assessed included usually two or three dishes (shown with pictures to the participants). I suggest that you show the list of all items in a table. Please also describe why you used more than one picture for each food category and why only the most liked option from each category was rated for pleasantness.

Validity of the findings

The raw data have been provided as an Excel file.

The authors should discuss how well the set of two/three figures showed to the participants represented the food category in question and how the selection of the set of figures may influence the results.

The results tell about food preferences of Polish women, but their connection to overweight was not confirmed (although the Introduction discusses overweight). It could be that pleasantness of food do not explain much of their use. Furthermore, self-reported responses are prone to bias from wish to provide responses that are socially acceptable. The authors could consider and discuss these challeges.

The Conclusions is a mere summary of the results; it does not really make conclusions based on the results.

Additional comments

Abstract includes too much details on results of statistical analysis, which makes it difficult for a reader to follow.

---

## Round 0.2 · Major Revisions

Thank you very much for your revision to the initial manuscript. The reviewers agree that many of the concerns previously raised have been addressed, but I tend to agree with the reviewer who has questioned if the analytical choices have falsely informed the conclusions offered. While I recognize that analytical styles vary, I believe the issues flagged are fundamental statistical issues.

While GLM is less sensitive to non-normal distributions than other models, that reduced sensitivity must be considered relative to the specific nature of the non-normal data--here, the outliers are an issue. There are many ways to address outliers. First, you can use a variety of statistical tools to identify how significant these outliers (e.g., Grubbs, Rosners) are and, if removed, how they affect the data and results. Second, handling statistically extreme outliers introduces bias from choice. You can take an approach, as recommended, to deal with them statistically, or to remove them. Third, you could also conduct the analysis showing the effect of the outliers on outcomes (analysis with and without) and determine how this would be usefully discussed in your paper (perhaps included as a supplement) and noted as a limitation of the study if the effect is great.

I also agree that the predictor variable has an abstract component (sensitivity) that has more than one concrete representations. The choice of representation is important when the analysis returns different outcomes depending on the representation, as suggested here.

Because the problems that would prevent the manuscript from being acceptable for publication can be overcome, I have returned the paper to you. Because your results from the new analysis will likely lead to different conclusions, I have marked the changes as major. If you disagree with the review of your analysis, I would encourage you to submit the paper elsewhere.

·

Basic reporting

Thank you for giving me the opportunity to review this revised version of the manuscript. It has improved a lot, especially textually. The additions are very useful, especially the pictures of the different food types, and the comment on social desirability in the Discussion section, for example.

With respect to the basic reporting, I still feel the discussion section is very long, and could be compressed more. Findings could be discussed in more general terms, reflecting the literature from the introduction, instead of separately for each predictor and food type. I leave it up to the editor to decide if the current length is acceptable for the journal.

Experimental design

As before, the design is correlational and correct.

Validity of the findings

Given the information in the manuscript, and the underlying data provided, I was able to re-analyse the data with the same results. That is very nice!

Regarding the validity of the findings, I still have a number of concerns. The authors have clarified many of the questions I had regarding the analyses (e.g., it is made clear that the analysis was a GLM, and hormonal status is clarified), but it remains questionable if the data supports the conclusions.

For example, the decision to keep strong outliers (i.e., someone with 60 pack years on a sample average of .8, SD = 4.7, so this person is approx. 12 standard deviations from the mean) in, raises doubt about the conclusions regarding smoking and food preference. It is possible, for example, that the reported relationship between smoking and disliking of bread is entirely due to this single person’s disliking of bread (nb. actually, when looking at the data, it is more likely removal of this individual makes the relationship between smoking and bread liking stronger).

Additionally, regarding the assumptions, the main assumption for regression is that there is a linear relationship between the predictor and dependent. If olfactory sensitivity entered the analysis in the format reported in Table 1 (or the data-file), i.e., the number of times the original concentration is diluted, this assumption is not met – in this form, olfactory sensitivity is neither normally distributed nor linearly associated with any of the dependent variables. If some transformation will take place (e.g., taking the natural log, or the dilution step), the analysis is probably valid.

I would find the manuscript publishable if the authors do the analyses again with the dilution steps as predictor instead of the raw value for number of times 59.9 PPM n-butanol was diluted as the operationalization for olfactory sensitivity, and take an approach to outliers (for example, replace using the Median Absolute Deviation method (Leys et al., 2013), or remove). This will affect the conclusions.

Leys, C., Ley, C., Klein, O., Bernard, P., & Licata, L. (2013). Detecting outliers: Do not use standard deviation around the mean, use absolute deviation around the median. Journal of experimental social psychology, 49(4), 764-766.

Additional comments

I don't have any general comments at this point, but a few additional specific comments, often in response to the rebuttal:
Lines 39-41: use M as abbreviation for averages (not x) – would further suggest to report SDs with to any means reported.
lines 199-201: I think my previous comment on people with obstructed nasal airflow, increased olfactory sensitivity and anosmia remains valid. My suggestion would be to rephrase this sentence as follows: "A transient nasal obstruction could excessively affect the results of the olfactory efficiency test, so the subjects with obstructed airflow could be classified as anosmic on the test day."
line 236: my suggestion would be to translate 'przyjemny' with 'pleasantness' and not with 'pleasure', for the purpose of the manuscript.
lines 250-252: it is not clear what these numbers are, there are no units. I think the numbers represent the times the n-butanol in the original concentration (59.9 ppm) is diluted, but this is not clear from the text.
Lines 285-289: the dependent variables, I think, are the rated pleasantness for each of these food types?
Line 352: olfactory memory?
Line 454: detection threshold.
Line 456: the standard deviation was, according to table 1, 15550 on a mean of 7456.
Table 1: a N-butanol dilution step mean of 7456 (SD = 15550) is not sensible and does not provide direct meaningful information about how sensitive the participants were for smells. This is the number of times the odor is diluted, not the dilution step. The median is step 10. There were 16 dilution steps, and the mean of these 16 should be used and reported.


N.B. these line numbers refer to the line numbers in the revised manuscript with tracked changes.

·

Basic reporting

The document has been improved in using clear and unambiguous professional English. The literature is still lacking on the interaction or association of hormone levels, taste and smell - the editions in lines 107 to 109 refers to menstrual cycle and calorie intake and body weight which is not the same as the link of hormone levels with taste and olfactory sensitivity. I suggest this is made clear the possible link between these three as this justifies the inclusion in the study whilst backing up the discussion which linked olfactory sensitivity to hormone levels.

Experimental design

A lot of improvements made to the methodology, and methods describe with sufficient detail and information to replicate.

Validity of the findings

No comment

Reviewer 3 ·

Basic reporting

This is OK. However, fluency of text could still be improved (see section 4).

Experimental design

The Authors have considered my comments and revised the manuscript accordingly.

Validity of the findings

The Authors have considered my comments and revised the manuscript accordingly.

Additional comments

The Authors have carefully considered my comments and revised the manuscript accordingly. I regard the manuscript much improved and have no further major concerns. However, fluency of the Abstract could still be improved. It is not clear what the values after the "x" are. Adding a note to Methods secotion of the Abstract that food preferences were measured using 9-point hedonic scale could help. In addition, "better sense of smell" is now ambiguous in "Women with a better sense of smell (shown in olfactometric measurements) preferred sour products, women with better smell (manifested as an ability to name scents) preferred sausages and ham." and only the extra info in the parentheses reveal the difference. I suggest to describe the differences directly, for example "Women performing better in olfactometric measurements preferred sour products, whereas women with better ability to name scents preferred sausages and ham."

---

## Round 0.3 · Minor Revisions

The final edits you made to the statistical analysis have strengthened the paper and I am happy to accept it. I am returning it to you for the one small change in language regarding the results. Once you submit, I can officially accept. Thanks so much for your attention and patience in this process!

·

Basic reporting

The manuscript has seen 3 rounds of revisions, and it has improved greatly. It is an interesting paper with interesting findings.

Experimental design

no further comments

Validity of the findings

I believe these findings presented are valid. The statistical analyses in the revised version seem sound.

Additional comments

The only suggestion I would like to add at this point is to rephrase the following type of sentence:
'Variances in broth preferences were explained in 13%, ... ' (e.g., in line 288, and the following lines). My suggestion would be to change it into 'The model for broth preferences explained 13% of the variance, ...'.

---

## Round 0.4 · accepted · Accept

Thanks for your attention to the final detail. I am pleased to accept your article!